# Risankizumab Therapy for Moderate-to-Severe Psoriasis—A Multi-Center, Long-Term, Real-Life Study from Poland

**DOI:** 10.3390/jcm12041675

**Published:** 2023-02-20

**Authors:** Michał Adamczyk, Joanna Bartosińska, Dorota Raczkiewicz, Kinga Adamska, Zygmunt Adamski, Maria Czubek, Beata Kręcisz, Elżbieta Kłujszo, Aleksandra Lesiak, Joanna Narbutt, Marcin Noweta, Agnieszka Owczarczyk-Saczonek, Witold Owczarek, Adam Reich, Dominik Samotij, Aleksandra Siekierko, Justyna Szczęch, Irena Walecka, Piotr Ciechanowicz, Anna Woźniacka, Agata Liszewska, Dorota Krasowska

**Affiliations:** 1Department of Dermatology, Venereology and Pediatric Dermatology Medical University of Lublin, 20-081 Lublin, Poland; 2Department of Cosmetology and Aesthetic Dermatology, Medical University of Lublin, 20-081 Lublin, Poland; 3Department of Medical Statistics, School of Public Health, Center of Postgraduate Medical Education, 01-826 Warsaw, Poland; 4Department of Dermatology, Poznan University of Medical Sciences, 61-701 Poznan, Poland; 5Departament of Dermatology Copernicus Sp z o.o., Podmiot Leczniczy Gdańsk, 80-803 Gdańsk, Poland; 6Department of Dermatology, Jan Kochanowski University of Kielce, 25-406 Kielce, Poland; 7Department of Pediatric Dermatology and Oncology, Medical University of Łódź, 92-215 Łódź, Poland; 8Department of Dermatology, Sexually Transmitted Diseases and Clinical Immunology, The University of Warmia and Mazury, 10-229 Olsztyn, Poland; 9Department of Dermatology, Military Institute of Medicine, 04-141 Warsaw, Poland; 10Department of Dermatology, Institute of Medical Sciences, Medical College of Rzeszow University, 35-055 Rzeszów, Poland; 11Centre of Postgraduate Medical Education, Dermatology Department, Central Clinical Hospital of the Ministry of the Interior and Administration, 02-507 Warsaw, Poland; 12Department of Dermatology and Venereology, Medical University of Łódź, 90-419 Łódź, Poland

**Keywords:** psoriasis, systemic treatment, biologic treatment, risankizumab, real-life experience

## Abstract

The present multi-center, long-term, real-life study made an attempt to assess the efficacy of risankizumab in the treatment of moderate-to-severe plaque psoriasis. The study comprised 185 patients from 10 Polish dermatologic departments undergoing risankizumab treatment. The disease severity was measured using the Psoriasis Area and Severity Index (PASI) before the start of the risankizumab treatment and next at the defined timepoints, i.e., 4, 16, 28, 40, 52 and 96 weeks of treatment. The percentage of patients achieving PASI90 and PASI100 responses as well as the PASI percentage decrease at the defined timepoints were calculated, and correlations with clinical characteristics and therapeutic effect were analyzed. The number of patients evaluated at the defined timepoints was: 136, 145, 100, 93, 62, and 22 at 4, 16, 28, 40, 52 and 96 weeks of treatment, respectively. At 4, 16, 28, 40, 52 and 96 weeks, the PASI90 response was achieved in 13.2%, 81.4%, 87.0%, 86.0%, 88.7% and 81.8% of patients, whereas the PASI100 response was achieved in 2.9%, 53.1%, 67.0%, 68.8%, 71.0% and 68.2% of patients, respectively. Our study revealed a significant negative correlation between a decrease in the PASI and the presence of psoriatic arthritis as well as the patient’s age and duration of psoriasis at several timepoints throughout the observation period.

## 1. Introduction

Psoriasis is a common, immune-mediated inflammatory disease that has considerable negative effect on patients’ quality of life and their daily functioning. At first understood merely as a skin condition, today, psoriasis is seen as a multisystem disease associated with numerous comorbidities, i.e., psoriatic arthritis (PsA), metabolic syndrome, atherosclerosis, cardiovascular disease (CVD) and depression [1]. While the majority of patients suffer from mild forms of psoriasis, with limited skin involvement treated with topical agents, the patients with more severe forms of the disease require more complex therapeutic strategies involving phototherapy and systemic pharmacotherapy, i.e., retinoids, immunosuppressants and biologic drugs.

With new biologic drugs on the market and their markedly improved efficacy, both doctors and patients expect prompt and long-lasting results of treatment which appear to be delivered via such biologics as risankizumab. Therefore, achieving PASI90 (Psoriasis Area and Severity Index), i.e., a 90% reduction in skin lesions expressed via the PASI, and PASI100, i.e., a 100% reduction in skin lesions expressed via the PASI is indicative of the high efficacy of biologic agents inhibiting IL-23 [2].

Risankizumab is a humanized monoclonal IgG1 antibody targeting the p19 subunit of IL-23. The efficacy and safety of risankizumab have been evaluated in several clinical trials, including long-term extension studies [3]. The initial phase three, double-blind placebo or active comparator-controlled studies, namely UltIMMa-1 and UltIMMa-2, demonstrated the excellent efficacy of risankizumab, especially in terms of achieving PASI90 and PASI100 [4]. The treatment was then prolonged into an ongoing long-term open-label extension trial (LIMMitless). An interim analysis of the risankizumab clinical trial confirmed its safety and efficacy through 176 weeks of therapy [5].

However, the results from clinical trials should be interpreted with caution mainly due to an inherent selection bias. Restrictive inclusion and exclusion criteria of many clinical trials narrow down the spectrum of enrolled individuals. Therefore, in many instances they may not be representative of the general population of patients with psoriasis. In contrast, real-life studies including subjects previously exposed to many different biologic drugs or having various comorbidities may provide valuable data not always available from clinical trials. Real-life data include case reports, case series and larger retrospective or prospective studies conducted by physicians in the clinical practice [6]. Comparison of real-life data with those derived from clinical trials may widen the knowledge about the efficacy and safety of different drugs, thus being very important in clinical practice.

The purpose of this study was to evaluate the efficacy of risankizumab in a real-life setting and compare our observations with the currently available data from real-life clinical studies. In our study, we also made an attempt to evaluate the effect of different clinical features of psoriatic patients on the efficacy of risankizumab.

## 2. Materials and Methods

### 2.1. The Study Group

Our study included patients previously qualified for risankizumab treatment in accordance with the requirements of The Polish Drug Program B.47, “Treatment of moderate to severe form of plaque psoriasis (ICD-10 L40.0)”. It comprised 185 adult patients suffering from moderate-to-severe plaque psoriasis (123 males (66.5%) and 62 females (33.5%)) who underwent risankizumab treatment in ten Polish dermatological clinics.

The inclusion criteria were as follows: severe psoriasis defined using a PASI ≥ 18, BSA ≥ 10 and DLQI ≥ 10 or patients with difficult-to-treat, severely affected sites (i.e., the scalp, face, palms, soles, anogenital region and nails), lack of response to treatment or contraindications to treatment with at least 2 conventional systemic drugs.

The exclusion criteria were as follows: hypersensitivity to risankizumab, severe, active infections (in particular tuberculosis), malignancies, active type B viral hepatitis, lymphoproliferative disorders, demyelinating diseases as well as pregnancy and breastfeeding.

A detailed medical history was taken, including age of psoriasis onset, duration of the disease, the presence of PsA and nail lesions, comorbidities (cardiovascular, metabolic, endocrine, hepatic, neurologic, psychiatric and gastrointestinal disorders), smoking status, family history of psoriasis and previous systemic and biologic treatment for psoriasis.

Prior to the treatment initiation, each study subject underwent physical examination, including measurement of body weight and height with calculation of body mass index (BMI). The severity of psoriasis was assessed at 0 weeks (initiation of treatment) as well as at 4, 16, 28, 40, 52, and 96 weeks. Disease severity was assessed using the PASI.

### 2.2. Risankizumab Administration and Dosage Regimen

Risankizumab was administered subcutaneously in the dose of 150 mg at 0 weeks, 4 weeks and then every 12 weeks thereafter. All the patients received at least one dose of risankizumab. At the time of the analysis, 136, 145, 100, 93, 62 and 22 patients completed 4, 16, 28, 40, 52 and 96 weeks of treatment, respectively.

### 2.3. Assessment of Risankizumab Efficacy

Risankizumab efficacy was assessed by means of the Psoriasis Area and Severity Index (PASI) performed before the start of risankizumab treatment, and next at the defined timepoints, i.e., 4, 16, 28, 40, 52 and 96 weeks of treatment.

Risankizumab efficacy was expressed using the PASI percentage decrease at the defined timepoints. Moreover, the percentage of patients achieving PASI90 and PASI100 responses was calculated. Correlations with clinical characteristics and therapeutic effect were analyzed. The number of patients evaluated at the defined timepoints was: 136, 145, 100, 93, 62 and 22 at 4, 16, 28, 40, 52 and 96 weeks of treatment, respectively.

As for safety, information about severe adverse events were collected.

### 2.4. Statistical Methods

All data were statistically analyzed using STATISTICA 13 software. Minimum and maximum values as well as means (M) and standard deviations (SD) were calculated for numerical variables, while absolute numbers (n) and percentages (%) of the occurrence of items for categorical variables.

The Student’s *t* test for paired data was used to compare PASI at 4, 16, 28, 40, 52 and 96 weeks with a baseline.

Pearson’s correlation coefficient was used to correlate the percentage decrease in the PASI at 4 weeks compared to the baseline with numerical characteristics. The same analyses were conducted at 16, 28, 40, 52, and 96 weeks. The Kruskal–Wallis H test was used to compare the percentage decrease in the PASI at 4 weeks compared to the baseline between patients with normal weight, overweight patients and obese patients. The same analyses were conducted at 16, 28, 40, 52 and 96 weeks. The Mann–Whitney U test was used to compare the percentage decrease in the PASI at 4 weeks compared to the baseline between two categories of categorical characteristics. The same analyses were conducted at 16, 28, 40, 52 and 96 weeks. The Student’s *t* test was used to compare the numerical characteristics between patients who achieved PASI90 and patients who did not achieve PASI90. The same analysis was conducted for PASI100. The Chi-square test was used to compare categorical characteristics between patients who achieved PASI90 and patients who did not achieve PASI90. The same analysis was conducted for PASI100. The significance level was set at *p* ≤ 0.05.

## 3. Results

### 3.1. Characteristics of the Study Subjects

Detailed clinical characteristics of the study subjects are presented in Table 1.

### 3.2. Changes in PASI Score

The mean changes in the PASI at the baseline (week 0) and weeks 4, 16, 28, 40, 52 and 96 are presented in Figure 1. At weeks 4, 16, 28, 40, 52 and 96, a significantly decreased PASI was observed in comparison to the baseline (*p* < 0.001). The percentage decrease in the PASI at weeks 4, 16, 28, 40, 52 and 96 is presented in Figure 2. At week 4, in 17.6% of patients the decrease in the PASI was between 60–70%, whereas in 13.2% of patients the decrease in the PASI was between 90–100% compared to the baseline values. At week 16, at least a 90% reduction in the PASI was observed in 81.4% of patients. At weeks 28, 40, 52 and 96, at least a 95% decrease in the PASI was observed in 81.0%, 79.6%, 82.3% and 77.3% of patients, respectively.

Figure 3 presents the percentage of patients who achieved PASI90 and PASI100 at weeks 4, 16, 28, 40, 52 and 96. The PASI90 response was observed in 13.2%, 81.4%, 87.0%, 86.0%, 88.7% and 81.8% of patients at weeks 4, 16, 28, 40, 52 and 96, respectively. The PASI100 response was observed in 2.9%, 53.1%, 67.0%, 68.8%, 71.0% and 68.2% of patients at weeks 4, 16, 28, 40, 52 and 96, respectively.

### 3.3. Correlations between PASI Changes and Selected Patient Characteristics

Table 2 presents correlations between the percentage decrease in the PASI at weeks 4, 16, 28, 40, 52 and 96 compared to the baseline and the characteristics of the study group. The decrease in PASI at each study timepoint did not correlate with age at psoriasis onset, weight, BMI, gender, smoking status, type I and II of psoriasis, cardiovascular disease and previous biologic therapy for psoriasis (*p* > 0.05).

A negative correlation was found between the patients’ age and the percentage decrease in the PASI at weeks 28, 40 and 52 compared to the baseline (r = −0.199, *p* = 0.047 at week 28; r = −0.310, *p* = 0.003 at week 40; r = −0.303, *p* = 0.017 at week 52). The older the age, the lower the decrease in the PASI.

A negative correlation was also revealed between duration of psoriasis and the percentage decrease in the PASI at weeks 40 and 52 compared to the baseline (r = −0.301, *p* = 0.003 at week 40; r = −0.317, *p* = 0.012 at week 52). The longer the duration of the disease, the lower the decrease in the PASI.

Compared to the baseline, at week 4 the patients with PsA had a lower percentage decrease in PASI than those without PsA (48.9 vs. 61.0, *p* = 0.025), and also at week 16 the patients with PsA showed a lower percentage decrease in PASI than those without PsA (91.8 vs. 94.8, *p* = 0.034).

At week 40, the patients without comorbidities had a higher percentage decrease in PASI than the patients with metabolic or hepatic disease (97.7 vs. 94.0, *p* = 0.006) and (97.6 vs. 92.7, *p* = 0.040), respectively (Table 2).

Table 3 presents correlations between PASI90 as well as PASI100 at week 16 and characteristics of the study group. The group of patients who achieved PASI90 at week 16 did not significantly differ from the group of patients who did not achieve it in terms of age, duration of psoriasis, age of psoriasis onset, weight, BMI, gender, smoking status, type I and II of psoriasis, cardiovascular, metabolic or hepatic diseases and previous biologic therapy (*p* > 0.05). A significantly lower percentage of patients with PsA (13/31, 41.9%) than patients without PsA (64/114, 56.1%) achieved PASI90 at week 16 (*p* = 0.028).

PASI100 at week 16 was achieved by patients with significantly lower BMI values (28.8 ± 5.7 kg/m^2^ vs. 31.2 ± 6.6 kg/m^2^ for patients who did not achieve it, *p* = 0.018) and lower body weight (88.6 ± 19.8 kg vs. 95.9 ± 21.7 kg, *p* = 0.038). The group of patients who achieved PASI100 at week 16 did not significantly differ from the group of patients who did not achieve it in terms of age, duration of psoriasis, age of psoriasis onset, gender, smoking status, type I and II of psoriasis, PsA, cardiovascular, metabolic or hepatic diseases and previous biologic therapy (*p* > 0.05).

Two severe adverse events were reported in the study group, i.e., the death of one patient not connected with risankizumab treatment and one pneumonia case requiring hospitalization.

## 4. Discussion

Risankizumab was evaluated in five phase III clinical trials (UltIMMa-1, UltIMMa-2, IMMvent, IMMhance, and IMMerge) [4,7,8,9], as well as in a long-term, open-label extension study (LIMMitless) [5]. All these trials provided data confirming both the excellent efficacy and safety of risankizumab which was maintained for at least 3 years of follow-up in the open-label treatment. Since risankizumab approval, real-life studies published to date have been providing evidence confirming its safety and efficacy [10,11,12,13,14,15,16]. In a large, retrospective, multinational, multicentric cohort study including a total of 3145 patients, after 18 months of therapy, the drug survival rate was the highest for risankizumab among all biologic therapies targeting IL-23 and IL-17 (96.4%; with 91.1% for guselkumab, 86.3% for brodalumab, 86.1% for ustekinumab, 82.0% for ixekizumab and 79.9% for secukinumab) [17]. The recently published extension of this study, comprising an even higher number of study subjects (*n* = 4178), showed similar results after 24 months of therapy [18]. The most important findings from real-life studies with risankizumab discussed below are summarized in Table 4.

In comparison to all the real-life studies on risankizumab effects published to date, our study consists of the highest number of study subjects, and the characteristics of our study population are similar to those from other real-life studies, including the data published by Mastorino et al. [14] Gkalpakiotis et al. [16] and Gargiulo et al. [10]. The fact that the real-life study of Gargiulo et al. [10] and ours are the longest, (104 and 96 weeks, respectively), they appear to provide reliable evidence confirming the long-term efficacy of risankizumab.

In clinical trials, PASI90 at week 16 was achieved by 73.8–75.3% of patients compared to 81.4% in our study [4,8]. In our patients, the mean PASI at week 16 was reduced from 21.12 to 1.13. It is worth noting that contrary to other real-life studies, in our study the percentage of patients achieving the PASI90 response at week 16 was higher than in clinical trials. This could be explained by the lower baseline PASI values in real-life studies than in clinical trials, and also by the fact that the real-life studies may include patients with limited skin involvement, so called “difficult-to-treat sites” (e.g., the scalp, palms and soles or anogenital region). As a matter of fact, the mean baseline PASI in our group was 21.12, which is almost the same as in UltIMMa-1 and UltIMMa-2 studies (20.6 and 20.5, respectively) [4]. On the other hand, i.e., Gargiulo et al. analyzed patients with a mean PASI of 13.52 and observed a lower PASI90 response rate (55.7%) at week 16 [10]. The same was observed by Mastorino et al., who found even lower PASI90 response rates (53%) but recruited subjects with a mean PASI of 12.5 ± 5.1 [14]. Importantly, the efficacy outcomes analyzed at later timepoints (weeks 40, 52 and 96/104) were much higher and more comparable to those from clinical trials (Table 4). For instance, in our study, at week 52, PASI90 and PASI100 was achieved in 88.7% and 71% of the study subjects, respectively, which is comparable to other studies. Treatment response was maintained until week 96, with PASI90 and PASI100 observed in 81.8% and 68.2% of the studied patients, respectively, which was almost the same as that noted by Gargiulo et al. at week 104 (80.8% and 69.2%, respectively) [10].

It is well known that high values of BMI have a negative effect on the efficacy of treatment with TNF-α antagonists, ustekinumab and secukinumab [19]. When it comes to the effect of BMI on Risankizumab therapy, the data from real-life studies bring contradictory results. However, a subgroup analysis of the patients participating in the IMMerge study revealed no impact of high BMI on response to risankizumab. Our analysis showed only one significant relationship: PASI100 at week 16 was achieved by patients with significantly lower BMI (28.8 ± 5.7 kg/m^2^ vs. 31.2 ± 6.6 kg/m^2^ for patients who did not achieve it, *p* = 0.018) and lower body weight (88.6 ± 19.8 kg vs. 95.9 ± 21.7 kg, *p* = 0.038). No differences were observed between the study subjects with normal BMI, those who were overweight and obese and decrease of PASI score at each study timepoint. Similarly, Gargiulo et al. [10] and Gkalpakiotis [16] found no correlation between BMI and response rates to risankizumab. On the contrary, Mastorino et al., Borroni et al. and Hansel et al. found a significant negative correlation between the BMI values and therapeutic effect of risankizumab [12,14,20]. Mastorino et al. reported that PASI100 at week 28 was achieved by only 22% of obese and 58% of non-obese patients (*p* = 0.007) [14]. Efficacy analysis performed by Borroni et al. showed a decrease in the chance of achieving a PASI90 response at week 40 by subjects with a higher BMI [12].

A previous exposure to biologic drugs may influence the efficacy of risankizumab. As risankizumab is a novel agent, it is frequently prescribed to patients that have been previously treated with one or more biologics. In our study, 38.4% of the studied patients were previously exposed to at least one biologic therapy and in our analysis, it did not matter if the patients had previously undergone biologic treatment, which was in agreement with Gkalpakiotis et al., Gargiulo et al. and Caldarola et al. [10,15,16]. Conversely, Mastorino et al. revealed that the patients previously treated with biologic drugs had lower response rates as expressed by the achievement of PASI90 and PASI75 at weeks 16 and 28 [14]. Hansel et al. revealed a significant difference in the achievement of PASI100 at weeks 36 and 52, [20] whereas Borroni et al. found a small difference in PASI75 rates at week 16 and a significant difference at week 40 [12]. Recent data published by Megna et al. show that risankizumab was effective and well tolerated in the population of psoriatic patients with a previous failure to anti IL-17 (secukinumab and ixekizumab), anti IL-12/23 (ustekinumab) and anti IL-23 (guselkumab) biologic drugs [13].

Our data support previous findings concerning the possible negative impact of the presence of PsA on the efficacy of risankizumab. In our study, 18.9% of patients were previously diagnosed with PsA. In the patients with PsA, we observed a lower percentage decrease in the PASI at weeks 4 and 16 in comparison to patients without PsA. Importantly, these differences were not observed at the consecutive timepoints of the study. Significantly poorer response rates in this subpopulation were observed by Mastorino et al., who found significantly higher mean PASI values in the studied PsA subgroup at all timepoints of their study in comparison to patients without PsA (2.7 vs. 1.7 (*p* = 0.036), 1.9 vs. 0.9 (*p* = 0.006)) and 4.1 vs. 0.5 (*p* = 0.016) at 16, 28 and 40 weeks, respectively) [14]. Nevertheless, other real-life studies did not confirm such a correlation.

With respect to comorbidities, Gargiulo et al. and Borroni et al. did not find any correlation between the presence of cardiometabolic diseases and response to risankizumab [10,12]. Interestingly, our analysis showed that patients with metabolic or hepatic diseases had a lower percentage decrease in PASI at week 40 than patients without such comorbidities.

Our analysis revealed an interesting correlation between the duration of psoriasis and response to risankizumab. The longer the duration of the disease, the lower the percentage reduction in the PASI at weeks 40 and 52. Our analysis also showed that the older the patient, the lower the decrease in the PASI at weeks 28, 40 and 52. It is thought that the earlier intervention with anti IL-23 biologic drugs may result in a better clinical response and more durable treatment effects after drug withdrawal. Such an effect may be due to the involvement of IL-23 in the differentiation and survival of tissue-resident memory T cells, which are thought to be responsible for recurrences of psoriasis at the previously affected sites after treatment cessation. Currently, there is an ongoing clinical trial with guselkumab (GUIDE study) which aims at evaluating the potential of this drug to modify psoriasis pathogenesis in subjects with early treatment intervention vs. those with long-lasting disease [21]. The results have not been published yet.

In our study, two severe adverse events were reported, i.e., the death of one patient who was not connected with risankizumab treatment, and another patient who required hospitalization because of pneumonia.

A multi-center character and a high number of participants may be considered as the strengths of the study. The limitations include its retrospective character and the lack of a control group. A relatively small number of subjects completed week 96, which may make the statistical analysis at this timepoint less accurate.

## 5. Conclusions

This study is the largest published to date real-life evidence of therapy with risankizumab. Our data confirm previous clinical trial and real-life data concerning the excellent effectiveness of risankizumab, also during a long-term, 96-week treatment. Several additional findings, including the negative impact of comorbidities, disease characteristics as well as previous treatments on the efficacy of risankizumab need to be confirmed in larger and preferably prospective studies. Real-life studies bring additional evidence from clinical practice that amend data derived from clinical trials.

## Figures and Tables

**Figure 1 jcm-12-01675-f001:**
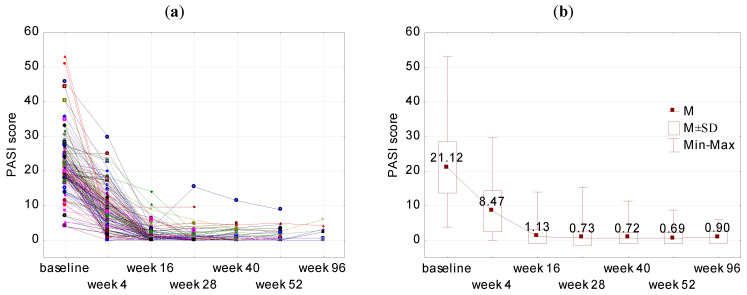
Changes in the PASI at weeks 4 (N = 136), 16 (N = 145), 28 (N = 100), 40 (N = 93), 52 (N = 62), 96 (N = 22). (**a**) PASI score in all study participants and (**b**) the mean PASI score with standard deviation (SD) during study timepoints. *p* < 0.001 for PASI at weeks 4, 16, 28, 40, 52 and 96 in all the comparisons to the baseline. *p* for Student’s *t* test for paired data.

**Figure 2 jcm-12-01675-f002:**
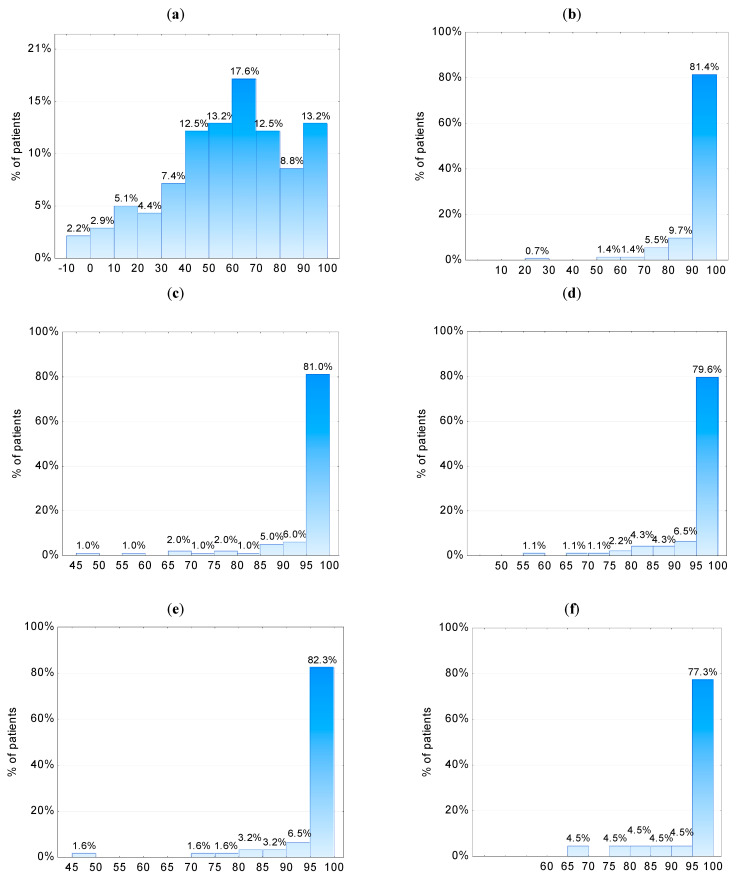
The percentage decrease in PASI at weeks (**a**) 4 (N = 136), (**b**) 16 (N = 145), (**c**) 28 (N = 100), (**d**) 40 (N = 93), (**e**) 52 (N = 62) and (**f**) 96 (N = 22) compared to the baseline.

**Figure 3 jcm-12-01675-f003:**
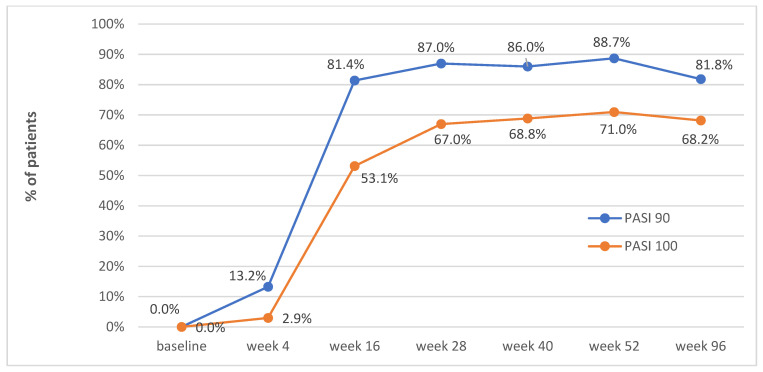
PASI90 and PASI100 at baseline (N = 185), weeks 4 (N = 136), 16 (N = 145), 28 (N = 100), 40 (N = 93), 52 (N = 62) and 96 (N = 22).

**Table 1 jcm-12-01675-t001:** Characteristics of the study subjects (N = 185).

Characteristics	Unit or Category	Results
Age, min–max, M ± SD	Years	19–83, 43.9 ± 14.2
Gender, *n* (%)	Male	123 (66.5)
	Female	62 (33.5)
Weight, min–max, M ± SD	kg	50–186, 91.1 ± 21.5
BMI, min–max, M ± SD	kg/m^2^	17.72–55.54, 29.75 ± 6.19
BMI groups, *n* (%)	Underweight	1 (0.5)
	Normal weight	38 (20.5)
	Overweight	68 (36.8)
	Obesity class I	44 (23.8)
	Obesity class II	23 (12.4)
	Obesity class III	11 (5.9)
Smoking status, *n* (%)	Non-smokers	119 (64.3)
	Ex-smokers	16 (8.6)
	Smokers	50 (27.0)
Psoriasis type, *n* (%)	I	166 (89.7)
	II	19 (10.3)
Family history, *n* (%)	Yes	74 (40.0)
Age of psoriasis onset, min–max, M ± SD	Years	3–69, 24.3 ± 12.7
Duration of psoriasis, min–max, M ± SD	Years	1–52, 19.7 ± 11.4
PASI, min–max, M ± SD		3.9–53.1, 21.12 ± 7.40
PsA, *n* (%)	Yes	35 (18.9)
Nail psoriasis, *n* (%)	Yes	84 (45.4)
Inverse psoriasis, *n* (%)	Yes	22 (11.9)
Comorbidities, *n* (%)	At least one	113 (61.1)
	Cardiovascular	70 (37.8)
	Metabolic	67 (36.2)
	Endocrine	25 (13.5)
	Hepatic	42 (22.7)
	Neurologic	6 (3.2)
	Psychiatric	17 (9.2)
	Gastrointestinal	23 (12.4)
Previous non-biologic treatment, *n* (%)	Methotrexate	146 (78.9)
	Cyclosporine A	120 (64.9)
	Retinoids	98 (53.0)
	Phototherapy	101 (54.6)
Previous biologic therapy, *n* (%)	Bio-naïve	114 (61.6)
	Bio-experienced	71 (38.4)
	Anti-TNF	46 (24.9)
	Anti-Il 17	18 (9.7)
	Anti-Il 12/23	17 (9.2)
	Anti-Il 23	12 (6.5)

**Table 2 jcm-12-01675-t002:** *p*-values for correlations of the percentage decrease in PASI at weeks 4, 16, 28, 40, 52 and 96 compared to baseline with characteristics of the study group.

Characteristics	Week 4	Week 16	Week 28	Week 40	Week 52	Week 96
Age (years) ^1^	0.708	0.559	**0.047**	**0.003**	**0.017**	0.660
Duration of psoriasis (years) ^1^	0.803	0.772	0.259	**0.003**	**0.012**	0.436
Age of psoriasis onset (years) ^1^	0.527	0.695	0.274	0.543	0.768	0.854
Weight (kg) ^1^	0.542	0.400	0.538	0.519	0.695	0.282
BMI (kg/m^2^) ^1^	0.670	0.478	0.999	0.411	0.647	0.272
BMI (normal weight vs. overweight vs. obesity) ^2^	0.438	0.582	0.598	0.175	0.234	0.991
Gender ^3^	0.457	0.149	0.151	0.778	0.120	0.999
Smoking status (smokers vs. non-smokers) ^3^	0.226	0.605	0.855	0.414	0.344	0.961
Psoriasis type (I vs. II) ^3^	0.408	0.198	0.854	0.451	0.763	0.369
PsA (yes vs. no) ^3^	**0.025**	**0.034**	0.232	0.057	0.081	0.349
Cardiovascular disease (yes vs. no) ^3^	0.088	0.066	0.304	0.966	0.699	0.282
Metabolic disease (yes vs. no) ^3^	0.585	0.111	0.098	**0.006**	0.078	0.536
Hepatic disease (yes vs. no) ^3^	0.861	0.242	0.104	**0.040**	0.152	0.755
Biologic therapy previously (yes vs. no) ^3^	0.283	0.134	0.188	0.123	0.482	0.319

^1^*p* for Pearson’s correlation coefficient between the percentage decrease in PASI at week 4 compared to the baseline and numerical characteristics. The same at weeks 16, 28, 40, 52 and 96. ^2^
*p* for Kruskal–Wallis H test to compare the percentage decrease in PASI at week 4 compared to the baseline between normal weight, overweight and obesity. The same at weeks 16, 28, 40, 52 and 96. ^3^
*p* for Mann–Whitney U test to compare the percentage decrease in PASI at week 4 compared to the baseline between two categories of categorical characteristics. The same at weeks 16, 28, 40, 52 and 96. Statistically significant correlations in bold.

**Table 3 jcm-12-01675-t003:** *p* for correlations of PASI90 and PASI100 at week 16 with characteristics of the study group (N = 145).

Characteristics	PASI90 (118 Patients Achieved PASI90 vs. 27 Patients Who Did Not Achieve PASI90)	PASI100 (77 Patients Achieved PASI100 vs. 68 Patients Who Did Not Achieve PASI100)
Age (years) ^1^	0.188	0.269
Duration of psoriasis (years) ^1^	0.229	0.239
Age of psoriasis onset (years) ^1^	0.734	0.905
Weight (kg) ^1^	0.164	**0.038**
BMI (kg/m^2^) ^1^	0.135	**0.018**
BMI (normal weight vs. overweight vs. obesity) ^2^	0.376	0.265
Gender ^2^	0.138	0.188
Smoking status (smokers vs. non-smokers) ^2^	0.331	0.947
Psoriasis type (I vs. II) ^2^	0.339	0.326
PsA (yes vs. no) ^2^	**0.028**	0.160
Cardiovascular disease ^2^	0.139	0.073
Metabolic disease ^2^	0.067	0.152
Hepatic disease ^2^	0.083	0.537
Previous biologic therapy ^2^	0.055	0.265

^1^*p* for Student’s *t* test to compare numerical characteristics between patients who achieved PASI90 and patients who did not achieve PASI90. The same for PASI100. ^2^
*p* for Chi-square test to compare categorical characteristics between patients who achieved PASI90 and patients who did not achieve PASI90. The same for PASI100. Statistically significant correlations in bold.

**Table 4 jcm-12-01675-t004:** PASI90 and PASI100 response percentage rates during different timepoints and impact of several clinical characteristics of patients on therapeutic response to risankizumab in real-life studies published to date, together with our study. Only studies including ≥50 subjects with treatment duration longer than 28 weeks were included.

Referrence, No of Subjects	Efficacy (PASI90/PASI100)					
**Week of Therapy**	**Week 4**	**Week 16**	**Week 28**	**Week 40**	**Week 52**	**Week 96/104**
Gkalpakiotis et al., *n* = 154 [16]	N/A	63.8/44.7	77.3/59.1	N/A	82.4/67.7	N/A
Gargiulo et al., *n* = 131 [10]	N/A	55.7/36.6	65.7/47.3	N/A	78.6/61.1	80.8/69.2
Graier et al., *n* = 55 [11]	N/A	63.9/42.6	70.0/50.0	N/A	72.0/60.0	N/A
Borroni et al., *n* = 66 [12]	N/A	61.0/28.6	N/A	85.5/62.3	N/A	N/A
Caldarola et al., *n* = 112 [15]	17.9/10.7	72.2/55.6	91.0/75.0	N/A	95.2/90.4	N/A
Mastorino et al., *n* = 166 [14]	N/A	53.0/32.0	72.0/51/0	73.0/53.0	82.0/73.0	N/A
Our study	13.2/2.9	81.4/53.1	87.0/67.0	86.0/68.8	88.7/71.0	81.8/68.2
**Disease/subject characteristics**	**BMI**	**Age**	**Duration of disease**	**Previous biologic therapy**	**Psoriatic arthritis**	**Cardiometabolic comorbidities**
Gkalpakiotis et al., *n* = 154	No impact	N/A	N/A	No impact	N/A	N/A
Gargiulo et al., *n* = 131	No impact	N/A	N/A	No impact	No impact	No impact
Borroni et al., *n* = 66	Lower response	N/A	N/A	Lower response	N/A	No impact
Caldarola et al., *n* = 112	No impact	Lower response	N/A	No impact	N/A	N/A
Mastorino et al., *n* = 166	Lower response	N/A	N/A	Lower response	Lower response	N/A
Our study	No impact	Lower response	Lower response	No impact	Lower response	Lower response

BMI—Body mass index, N/A—not assessed.

## Data Availability

All data presented in this study are reported in this manuscript.

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
