# Peer review of "Risankizumab Therapy for Moderate-to-Severe Psoriasis—A Multi-Center, Long-Term, Real-Life Study from Poland"

_jcm, 2023, doi:10.3390/jcm12041675_

Round 1

Reviewer 1 Report

Adamczyk et al present a real-life observation of 185 psoriasis patients treated with the IL-23 p19 antibody risankizumab over a period of 96 weeks. Primary outcome parameters were the achievment of PASI 100 and PASI 90 response. The results of the study widely confirm the data from published phase 3 and 4 studies with risankizumab. The methods are described adequately, the data are well presented and convincing, and subanalyses interesting. 

I have only minor suggestions to improve the paper. I.e. the comparison between short-term disease duration (STD) and long-term disease duration (LTD) patients is of great interest. Studies with the p19 antibody guselkumab and experimental data have shown that STD patients respond more rapidly to IL-23 inhibition, and early treatment with IL-23 antibody may prevent chronification of psoriasis. It is thought that early intervention may prevent the establishment of tissue resident memory cells (TRM) that drive and perpetuate the chronification of psoriasis. The authors of the study presented here confirm the more rapid response in STD patients. This observation is worth more extensive discussion and citation of corresponding literature.

Author Response

Dear Reviewer,

Thank you very much for considering our work interesting and worth publication in JCM. We are very pleased with your valuable comments and suggestions for changes in manuscript. Please be kindly informed, that we performed changes as requested by you and the rest of reviewers. We have added a part in discussion about potential of IL-23 inhibitors to affect psorriasis pathogenesis and cited study protocol, which evaluates potential role of guselkumab in modifying pathogenesis of psoriasis. Unfortunetely no results have been published yet,and thus are not available to refer to.

With best regards,

The Authors.

Reviewer 2 Report

I think it is always helpful to implement real-life experiences in the literature, especially from multiple centers and with a substantial number of patients, so the paper has some potential; however, it is very verbose, the same results could be reported in a much more concise article , with fewer tables and figures; The report of real life experiences is useful, but the table should be more intuitive at a glance.

More in detail:

1) Materials and methods section needs to be improved and detailed: what were the inclusion and exclusion criteria? how many risankizumab administrations were needed to enter the study? how were the patients who dropped out of the study evaluated? and which features did they have? 

2) Results section: data is redundant as it is also reported in the tables and figures , the section needs to be simplified a lot 

3) Figures: Figures 1, 2 and 4 are not very illustrative of the results, I would only keep figure 3 

4) the discussion is very long-winded 

Author Response

Dear Reviewer,

Thank you very much for considering our work interesting and worth publication in JCM. We are very pleased with your valuable comments and suggestions for changes in manuscript. Please be kindly informed, that we performed changes as requested by you and the rest of reviewers. Below, please find responses for your comments:

1) Materials and methods section needs to be improved and detailed: what were the inclusion and exclusion criteria? how many risankizumab administrations were needed to enter the study? how were the patients who dropped out of the study evaluated? and which features did they have? – The section was divided into 4 subsctions, and more details were given, we included inclusion and exclusion criteria as requested. All patients received at least one dose of risankizumab. No patients dropped out of the study. 

2) Results section: data is redundant as it is also reported in the tables and figures , the section needs to be simplified a lot – we removed some informations from the text to simplify the results section.

3) Figures: Figures 1, 2 and 4 are not very illustrative of the results, I would only keep figure 3 – dear Reviewer, after carefull consederation of this comment, we decided to remove parts c-f of figure 1 and Figure 2. However, we think, that parts a and b from figure 1, as well as figure 2 should remain in the article, as they present important findings described in the article. We kindly ask for permission to leave them in manuscript.

4) the discussion is very long-winded – We have shortened this part of publication.

With best regards,

The Authors.

Reviewer 3 Report

The authors reported the results of a multi-center, long-term, real-life trial investigating the efficacy and safety of risankizumab in a real-life setting. 

The manuscript is interesting and well-written. However, there are minor concerns.

My comments:

- English language should be revised.

- Abstract: abstract should be unstructured and submission guidelines should be followed (https://www.mdpi.com/journal/jcm/instructions#preparation)

- Introduction: more data on the difference between clinical trials and real-life should be discussed. You should read and cite "Cinelli E, et al. Real-world experience versus clinical trials: pros and cons in psoriasis therapy evaluation. Int J Dermatol. 2022;61(3):e107-e108. doi:10.1111/ijd.15644"

- Introduction: before using the abbreviations (e.g. PASI) you should report the word you are abbreviating 

- Introduction: please explain the significance of PASI75 and PASI90

- Material and Methods: exclusion criteria should be discussed

- Material and Methods: please discuss safety assessment

- Results: I really appreciated this section of the manuscript

- Results: data about the safety should be discussed

- Discussion: in the review of the literature investigating the effectiveness and safety of risankizumab in real-life you have missed this manuscript "Megna M, et al. Risankizumab treatment in psoriasis patients who failed anti-IL17: A 52-week real-life study. Dermatol Ther. 2022;35(7):e15524"

- Discussion: strengths and limitations should be discussed

Author Response

Dear Reviewer,

Thank you very much for considering our work interesting and worth publication in JCM. We are very pleased with your valuable comments and suggestions for changes in manuscript. Please be kindly informed, that we performed changes as requested by you and the rest of reviewers. Below, please find short responses to your comments:

- English language should be revised. – The article was carefully read again and corrected.

- Abstract: abstract should be unstructured and submission guidelines should be followed (https://www.mdpi.com/journal/jcm/instructions#preparation) – Abstract was corrected as per  journal guidelines.

- Introduction: more data on the difference between clinical trials and real-life should be discussed. You should read and cite "Cinelli E, et al. Real-world experience versus clinical trials: pros and cons in psoriasis therapy evaluation. Int J Dermatol. 2022;61(3):e107-e108. doi:10.1111/ijd.15644"

- Introduction: before using the abbreviations (e.g. PASI) you should report the word you are abbreviating – This was corrected as requested, thank you for paying attention on that.

- Introduction: please explain the significance of PASI75 and PASI90 – A sentence about subject’s expectations added toIntrduction section.

- Material and Methods: exclusion criteria should be discussed – Added as per request.

- Material and Methods: please discuss safety assessment –  Data about SAEs during study included into manuscript as requested.

- Results: I really appreciated this section of the manuscript

- Results: data about the safety should be discussed – Data about SAEs during study included into manuscript as requested.

- Discussion: in the review of the literature investigating the effectiveness and safety of risankizumab in real-life you have missed this manuscript "Megna M, et al. Risankizumab treatment in psoriasis patients who failed anti-IL17: A 52-week real-life study. Dermatol Ther. 2022;35(7):e15524" Thank you for paying attention on that, we have found and cited this reference in Discussion section,

- Discussion: strengths and limitations should be discussed – the last paragraph of discussion have been expanded, as requested.

With best regards,

The Authors.

Round 2

Reviewer 2 Report

the main changes have been made, the article in my opinion is now suitable for publication, after language revision 

Reviewer 3 Report

All the changes have been made. The manuscript is now suitable for publication.